# Terrestrial Water Loss at Night: Global Relevance from Observations and Climate Models

Ryan S. Padrón[1], Lukas Gudmundsson[1], Dominik Michel[1], Sonia I. Seneviratne[1]

[1]Institute for Atmospheric and Climate Science, Department of Environmental Systems Science, ETH Zurich, Zurich, 8092, Switzerland

*Correspondence*: Ryan S. Padrón (ryan.padron@env.ethz.ch)

**Abstract.** Nocturnal water loss (NWL) from the surface into the atmosphere is often overlooked because of the absence of solar radiation to drive evapotranspiration and the measuring difficulties involved. However, growing evidence suggests that NWL – and particularly nocturnal transpiration – represents a considerable fraction of the daily values. Here we provide a global overview of the characteristics of NWL based on latent heat flux estimates from the FLUXNET2015 dataset, as well as from simulations of global climate models. Eddy-covariance measurements at 99 sites indicate that on average NWL represents 6.3 % of total evapotranspiration. There are six sites where NWL is higher than 15 %; these are mountain forests with considerable NWL during winter related to snowy and windy conditions. Higher temperature, vapor pressure deficit, wind speed, soil moisture and downward longwave radiation are related to higher NWL, although this is not consistent across all sites. On the other hand, the global multi-model mean of terrestrial NWL is 7.9 % of total evapotranspiration. The spread of the model ensemble, however, is greater than 15.8 % over half of the land grid cells. Finally, NWL is projected to increase everywhere with an average of 1.8 %, although with a substantial inter-model spread. Changes in NWL contribute substantially to projected changes in total evapotranspiration. Overall, this study highlights the relevance of water loss during the night and opens avenues to explore its influence on the water cycle and the climate system under present and future conditions.

## 1 Introduction

Water is lost from the surface to the atmosphere through evapotranspiration (ET). This process interlinks the water, energy and carbon cycles, and hence influences climate, ecology, agriculture, and economy (e.g. Betts et al., 1996; Fisher et al., 2017; Zhang et al., 2015). Although daytime ET, mainly driven by solar radiation, represents the majority of the contribution to total water loss, nighttime ET is likely non negligible. Nocturnal water loss may occur as evaporation from soil and canopy, snow sublimation, or plant transpiration through stomatal and cuticular conductance. It is also recognized that vapor pressure deficit, temperature, wind speed, longwave radiation and surface resistance influence nocturnal ET (Monteith, 1965; Penman, 1948). The prevalence of nocturnal water loss and its significance for the surface water and energy balance, however, remains overlooked and unclear.

In recent years there has been a growing body of evidence about the occurrence of nocturnal ET, with a specific focus on transpiration (Tr). Observations of nocturnal stomatal conductance across hundreds of species have challenged the assumption of stomatal closure in the absence of photosynthetically active radiation (e.g. Daley and Phillips, 2006; Dawson et al., 2007; Lombardozzi et al. 2017; Snyder et al., 2003). Possible advantages of nocturnal sap flow include capacitance refilling, embolism removal, nutrient uptake, hydraulic redistribution and oxygen supply (Zeppel et al., 2014), whereas it remains unclear if Tr with no associated carbon gain has any benefits for vegetation or is simply unavoidable. Total water loss through ET, however, is more relevant than Tr from a water balance perspective since it additionally includes evaporation or snow sublimation from the ground and canopy. Nocturnal ET can be measured with lysimeters or eddy-covariance (EC) flux systems. A summary of previously reported nocturnal water loss estimates of both Tr and ET is provided in Table 1.

**Table 1.** Nocturnal transpiration (Tr) and evapotranspiration estimates (ET) reported in the literature.

| Nocturnal water loss | Measurement type | Vegetation type | Setup | Location | Reference |
|---|---|---|---|---|---|
| Tr (rate): 5–15 % of daytime rates typically, max: 30 % | Porometer, gas exchange, sap flow, lysimeter | Multiple $C_3$ and $C_4$ species | Field, lab, growth chamber, greenhouse | Multiple | Caird et al. (2007) |
| Tr: 10–25 % of total | Estimate from published literature | Typical plant functional types | Not available | Not available | Zeppel et al. (2014) |
| ET (annual): 3.5–9.5 % of daytime total | Lysimeter | Grass (plus shrub and moss) | Field | Western Germany | Groh et al. (2019) |
| ET: 12–23 % of daytime total | Lysimeter | Bean and cotton row-crops | Ecotron: controlled conditions | Montpellier (France) | de Dios et al. (2015) |
| ET: 6 % of total | Eddy-covariance | Oak - grass savanna | Field | California (US) | Fisher et al. (2007) |
| ET: 1 % of total | Eddy-covariance | Pinus Ponderosa forest | Field | California (US) | Fisher et al. (2007) |
| ET: 8–9 % of daytime total | Eddy-covariance | Grass field, Pine plantation, and hardwood forest | Field | Co-located sites in North Carolina (US) | Novick et al. (2009) |

Water is not only lost from the surface during night, but it can also be gained by dew formation. For example, dew and hoar frost amounts to 4.2–6.4 % of annual precipitation in three humid grass sites in Austria and Germany (Groh et al., 2018, 2019), and was found to occur in approximately 30 % of the nights in a forest in central Colorado (Berkelhammer et al., 2013) and 70 % of the nights in a grassland in the Netherlands (Jacobs et al., 2006). ET and dew formation correspond to a latent heat flux and might both occur for example within the same hour, resulting in difficulties to disentangle them if the temporal resolution of the data is insufficient. In the present study, we therefore focus on the net latent heat flux or net nocturnal water loss (NWL) defined as ET minus dew formation.

Climate models generally represent latent heat flux as a function of the air-surface gradient in specific humidity and a resistance to water vapor transfer. This total resistance can include an aerodynamic resistance, a resistance to diffusion

through the soil, a leaf boundary layer resistance and stomatal resistance. Stomatal resistance or conductance is parameterized in most large-scale land surface models similarly to the Ball–Woodrow–Berry model (Ball et al., 1987; Ball, 1988; Collatz et al., 1991; Leuning, 1995; Medlyn et al., 2011; Sellers et al., 1996), i.e. as a linear function where the intercept is assumed to represent nocturnal conductance (see explanation in Lombardozzi et al., 2017). Meanwhile, new
evidence suggests that nocturnal stomatal conductance is actively controlled, and that it is not equivalent to minimum conductance (Duursma et al., 2019). Underestimation of nocturnal stomatal conductance would lead to lower transpiration, and hence lower NWL. Previous research has noted that land surface models, dynamic global vegetation models and ecophysiological models continue to commonly assume that virtually no transpiration takes place at night, despite evidence suggesting otherwise (e.g. Lombardozzi et al., 2017; Zeppel et al., 2014). By adjusting the nocturnal stomatal conductance of
the Community Land Model (CLM) version 4.5 based on empirical evidence, Lombardozzi et al. (2017) obtain an increase of up to 5 % in global transpiration, as well as significant effects on soil moisture availability and carbon uptake. In another study, Vinukollu et al. (2011) reported a mean nocturnal ET from the VIC land surface model of 9.6 % relative to daytime ET. It is also known that simple land evaporation models are not well suited for nocturnal conditions (Ershadi et al., 2014). Finally, to our knowledge, there have not been any studies analyzing NWL estimates from an ensemble of global climate
models.

The goal of this study is to provide an overview of the magnitude and variability of NWL across the globe, as well as to explore its relationship to different meteorological and land cover conditions. An improved understanding of this overlooked flux is relevant for the surface water and energy balance. Until now most research about NWL stems from the plant
physiology community, whereas the relevance of their results for hydrological and climate studies is yet to be fully explored. Here we analyze observations of NWL from a lysimeter and a global network of EC measurements, together with estimates from a climate model ensemble for present and projected future conditions. We conclude with a comparison of the observed and modeled data, while keeping in mind the difference in spatial resolution.

## 2 Data

### 2.1 Observations

### 2.1.1 Co-located lysimeter and EC station

Water fluxes are measured by a co-located weighing lysimeter and EC tower (2 m height) at the Rietholzbach pre-alpine catchment in Northeastern Switzerland (47.38° N, 8.99° E; 795 m a.s.l.; see Seneviratne et al., 2012 for site details). The sensors are thoroughly described by Hirschi et al. (2017). Given that in this case the focus is on sensor comparison, day and
night are distinguished using a simple threshold of 10 W m$^{-2}$ for measured incoming solar radiation below which it is

assumed that no photosynthesis occurs (Hirschi et al., 2017). Data from 2010 to 2018 are used for comparing NWL estimates from these two independent measurement techniques.

For the lysimeter, changes in the total system mass (i.e. its weight plus accumulated seepage) are quantified every 5 minutes and correspond to water lost as ET or gained by precipitation, including dew. We apply an adaptive window and adaptive threshold (AWAT) filter to the total system mass of the lysimeter to reduce noise in the timeseries (Peters et al., 2014; Ruth et al., 2018). A minimum of 5 minutes and maximum of 45 minutes are assumed for the moving-average window, as well as a minimum of 0.01 mm and a maximum of 0.25 mm for the threshold values to distinguish signal from noise. A piecewise cubic Hermitian spline is used to interpolate between points of significant mass change (Peters et al., 2016), after applying an 85$^{th}$ percentile "snap routine" at inflection points (Peters et al., 2017). We estimate dew formation from hourly weight increases in the lysimeter when a co-located rain gauge does not record precipitation in that hour or the next. Note that very light precipitation might not be recorded due to the 0.1 mm rain gauge resolution. In those rare occasions when estimated dew surpasses a maximum formation rate of 0.07 mm h$^{-1}$ (Monteith and Unsworth, 1990), it is instead attributed as rain or snow. NWL is calculated as ET minus dew. Lysimeter data from December to March are discarded because the quality is strongly affected by formation of snow bridges and the occurrence of snow drift. In addition, data from the following months are also omitted due to cases with unrealistic lysimeter weight and/or seepage measurements: July–September 2017, August 2014 and 2016, and November 2010, 2011 and 2016.

The EC data are processed with EddyPro (Fratini and Mauder, 2014; LI-COR, 2018) to obtain a latent heat flux time series with a temporal resolution of 30 minutes. Values are discarded for intervals when rain occurs, when the tower is in the upwind direction affecting the air flow (see Hirschi et al., 2017), and for cases with too low turbulence (median threshold for friction velocity) based on Wutzler et al. (2018). The resulting gaps are filled according to Reichstein et al. (2005). Latent heat flux is converted into water volume by dividing over the latent heat of vaporization; here we assume $\lambda = 2.472E6$ J kg$^{-1}$.

### 2.1.2 Global network of EC stations

To obtain a broader picture of NWL across the globe we employ the FLUXNET2015 Tier 1 dataset, which provides EC measurements of latent heat flux together with numerous other meteorological variables from a global network of 166 sites. We further select only those stations that contain at least 3 years of data to obtain a more accurate climatology of NWL. The temporal resolution of the data is 30 minutes. There are implemented tailored steps for quality assurance and quality control (Pastorello et al., 2014). A quality flag at each time interval indicates whether the data were measured or gap-filled based on marginal distribution sampling (Reichstein et al., 2005). Moreover, there is an energy balance closure correction factor applied to the data based on the assumption that the Bowen ratio is correct. A *joint uncertainty* estimate that combines a *random uncertainty* component and an energy balance closure component is provided at each timestep. Full details of the data processing are available at https://fluxnet.fluxdata.org/data/fluxnet2015-dataset/data-processing/. Even though the

dataset distinguishes between daytime and nighttime intervals based on potential incoming solar radiation, we additionally determine the total number of nighttime hours by calculating the sunset and sunrise time of each day (see https://www.esrl.noaa.gov/gmd/grad/solcalc/calcdetails.html). Finally, this study uses data from 99 sites (see Table A1) that include energy balance corrected measurements of latent heat flux, as well as the uncorrected fluxes.

Here we assume that the provided uncertainty for latent heat flux at each timestep $i$ is the standard deviation ($\sigma_i$) of a normal distribution, and thus propagate it to obtain the uncertainty of the accumulated flux ($\sigma_{sum}$) over $n$ timesteps as follows:

$$\sigma_{sum} = \left( \sum_{i=1}^{n} \sigma_i^2 + \sum_{j=1}^{n-1} \sum_{k=j+1}^{n} 2\rho_{jk}\sigma_j\sigma_k \right)^{0.5} \qquad (1),$$

where $\rho_{jk}$ corresponds to the Pearson correlation between the estimates of timesteps $j$ and $k$. Because there is no information available to compute this correlation, we assume an average $\rho_{jk} = 0$ in accordance with the FLUXNET2015 data processing. In addition, note that EC measurements do not account for latent heat storage in the air between the ground and measurement level. Lastly, it is important to be aware that the reliability of EC measurements decreases during the night due to low and intermittent turbulence (e.g. Baldocchi, 2003; Moffat et al., 2007). Nonetheless, on average across all analyzed sites, latent heat flux is measured in 60 % of all nighttime intervals, whereas gap-filling is required in the remaining 40 %.

## 2.2 Climate models

Sub-daily climate model output is required to study NWL. Here we analyze an ensemble of climate model simulations of the fifth phase of the Coupled Model Inter-comparison Project (CMIP5) that provide 3 hourly estimates of latent heat flux. As for the EC data, we obtain NWL by dividing it over the latent heat of vaporization λ. For present conditions we use data from historical simulations during the period 1976–2005, whereas for the future period 2081–2100, we use data from simulations with the "business as usual" RCP8.5 emissions scenario (Moss et al., 2010). The employed ensemble comprises 26 different models (or model configurations) with one initial condition simulation (see Table A2). Data from all models are bilinearly interpolated to a common 2.5° × 2.5° grid. Grid cells with data from less than 2/3 of all models are not considered.

To estimate total NWL we obtain the average flux from all 3 hourly intervals that are exclusively night, and then extrapolate this value based on the complete number of nocturnal hours. To achieve this, we compute the time of sunset and sunrise for each day at the center of each individual grid cell using the solar time equations without accounting for topography. Note that this extrapolation approach could lead to inaccuracies if the NWL rate from periods immediately following sunset or just prior to sunrise systematically differ from the NWL rate during the middle of the night.

# 3 Results

## 3.1 Observed nocturnal water loss

Monthly NWL from the co-located lysimeter and EC system show a Pearson correlation of 0.5 or 0.57, depending on how dew is estimated from the lysimeter data (L1 vs. L2, see Figs. 1a and 1b). For L1 (Fig. 1a), the default threshold of 0.07 mm

h$^{-1}$ is used (Section 2.1.1). In the case of L2 (Fig. 1b), we select here as a sensitivity test a second threshold of 0.035 mm h$^{-1}$, i.e. half of the defined value of 0.07 mm h$^{-1}$ for maximum dew formation, when processing the lysimeter data. Note that the correlations may be affected by the difference in the footprint of the sensors and periods with gap-filled EC data. Also, in this case there is no energy balance closure correction factor applied to the EC data. The agreement between EC and lysimeter improves if the NWL monthly climatology is analyzed. Moreover, in months when one of the lysimeter estimates

of NWL is either too high or too low relative to the EC data, the other lysimeter estimate generally has a much better agreement. Overall, these results suggest that EC measurements can provide meaningful estimates of NWL. The annual climatology of EC-based NWL at this particular grassland site in Switzerland is 34.3 mm, equivalent to 5.8 % of annual ET.

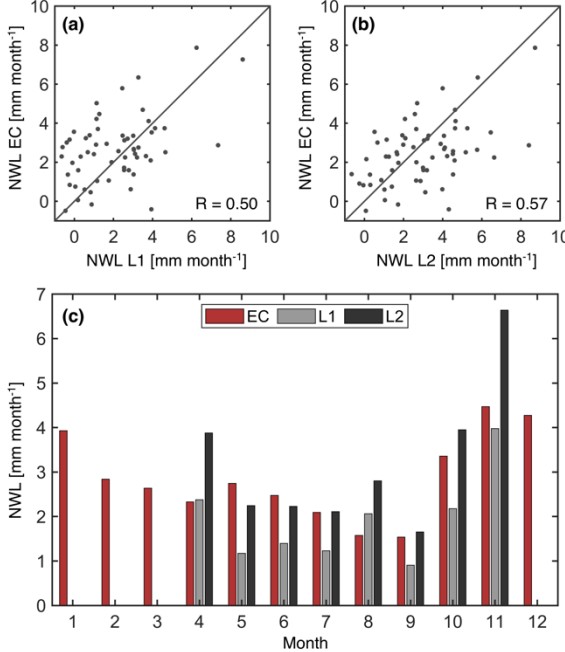

**Figure 1.** Comparison of nocturnal water loss (NWL) measured by the co-located lysimeter and EC system at Rietholzbach. Comparison
of individual months is shown in (a) and (b) with the Pearson correlation coefficient denoted as R, whereas a comparison of the climatology from the period 2010–2018 is shown in (c). L1 corresponds to the lysimeter estimate with a maximum dew formation threshold of 0.07 mm h$^{-1}$, and L2 with a threshold of 0.035 mm h$^{-1}$. Lysimeter data from December to March are discarded because of measurements issues when snow is present.

An overview of observed NWL at the analyzed FLUXNET sites is presented in Fig. 2. Mean annual NWL based on energy

balance corrected fluxes is 44.2 mm on average over all 99 stations, whereas the 5[th] and 95[th] percentiles of the distribution are 4.5 mm and 140.9 mm. There is a positive Spearman correlation coefficient of 0.61 between total ET and NWL,

indicating generally higher NWL at sites with higher ET. The net nocturnal water loss as a fraction of total ET, i.e. $NWL_f$ = NWL / ET, provides more insight on the relevance of the nocturnal water flux. Average $NWL_f$ across all stations is 6.3 %, the 5th percentile is 1 %, and the 95th percentile is 15.6 %. These annual mean values are computed from monthly climatologies obtained by omitting months with half or more of missing latent heat flux data. There is practically no difference in the distribution of $NWL_f$ with and without energy balance closure correction, whereas NWL is generally smaller when based on uncorrected fluxes. Furthermore, the uncertainty of annual mean $NWL_f$ per site, given by $2\sigma$ (~95 % confidence interval), is rather small with an average of ± 0.15 %. When assuming a more conservative value of $\rho_{jk}$ = 0.1 in equation 1, the average uncertainty across sites increases to ± 1.7 %.

Interannual variability of $NWL_f$, represented by the standard deviation, is 2.4 % on average from all sites. To analyze seasonality, we compute NWL for the trimesters December–February (DJF), March–May (MAM), June–August (JJA) and September–November (SON) at all 81 sites located above 30° N, where seasonal differences are clearer, and data are available. The most common season with the highest NWL is winter (35.8 % of the sites) followed by autumn (25.9 %), summer (23.5 %) and spring (14.8 %); whereas for the lowest NWL, the most common is summer (37 %) and the least common is autumn (13.6 %). Note that this is partly related to an increase in the total nocturnal hours as we go from summer to autumn and winter.

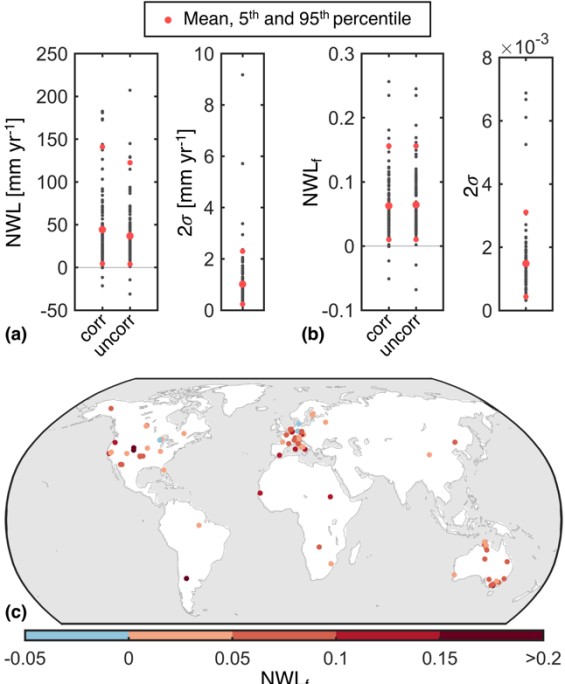

**Figure 2.** Nocturnal water loss at 99 FLUXNET sites as the annual NWL (a), and as the fraction of total evapotranspiration $NWL_f$ (b). Values from individual sites are shown in black, whereas the mean, 5th percentile and 95th percentile are shown in red. Both energy balance corrected values (corr) and uncorrected values (uncorr) are shown. Uncertainty estimates are given by $2\sigma$, which correspond to a

confidence interval of approximately 95 %. The uncertainty of total ET is small and therefore neglected when computing the uncertainty of $NWL_f$. (c) Location of sites with their estimated $NWL_f$.

The variability in $NWL_f$ across sites cannot be easily explained by annual average climate conditions (temperature and precipitation) or land cover (Fig. 3). Nonetheless, deciduous broadleaf forests (DBF) have an overall lower $NWL_f$, whereas

evergreen needleleaf forests (ENF) include most cases with higher $NWL_f$. An ANOVA test (differences in the mean) for the land cover categories has a p-value of 0.038, and a Kruskall-Wallis test (differences in the distribution) a p-value of 0.055. The three sites with negative $NWL_f$ (dew is greater than nocturnal ET) are Hainich (Germany), Soroe (Denmark), and Willow Creek (WI, USA). These are all DBF with typically lower vapor pressure deficit and higher soil moisture than approximately 75 % of all sites. Moreover, it may be more difficult to accurately measure EC latent heat flux at DBF sites

with large trees that reduce the ground-atmosphere coupling. On the other hand, there are six sites with $NWL_f > 15$ %: GLEES (WY, USA), GLEES Brooklyn tower (WY, USA), Niwot Ridge Forest (CO, USA), Lavarone (Italy), Wallaby Creek (Australia), and San Luis (Argentina). These are four ENF, an evergreen broadleaf forest (EBF) and a mixed forest (MF) in mountainous areas. Winter contribution to annual NWL approximately doubles that of summer in the four ENF sites. Snowier and windier conditions at these sites may suggest a considerable contribution of sublimation to NWL. The

percentage of gap-filled data for these sites with relatively high or low NWL is not particularly different than for all other sites.

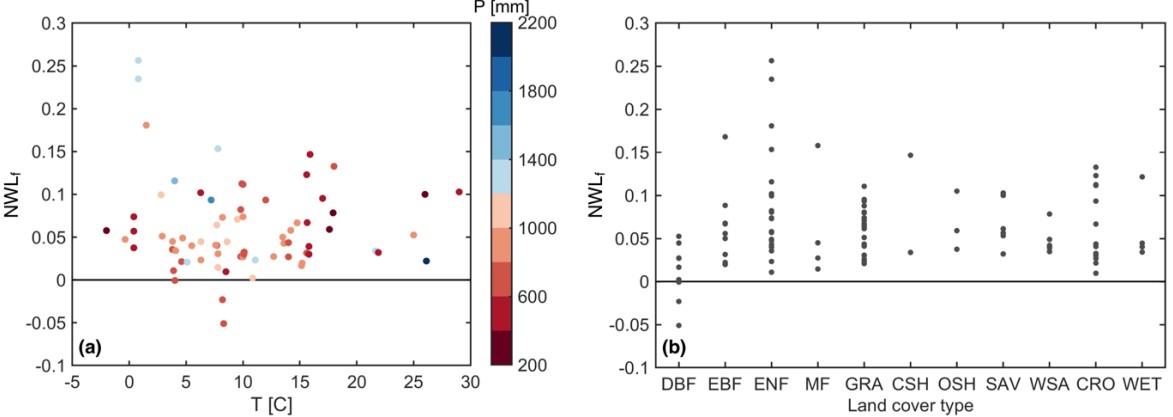

**Figure 3.** Relation of $NWL_f$ with (a) mean annual temperature (T) and precipitation (P), and with (b) land cover type at FLUXNET sites. Precipitation and temperature data are available for 73 of the 99 FLUXNET sites. Land cover types are deciduous broadleaf forest (DBF),
evergreen needleleaf forest (ENF), evergreen broadleaf forest (EBF), mixed forest (MF), grassland (GRA), closed shrubland (CSH), open shrubland (OSH), savanna (SAV), woody savanna (WSA), cropland (CRO), and wetland (WET).

At most sites there is a positive correlation of NWL with local air temperature (T), vapor pressure deficit (VPD), wind speed (WS), soil moisture (SM) and downward longwave radiation (LWd) for the 30-minute non-gap-filled data (Fig. 4). Correlations with net radiation (Rn) and ground heat flux (G) are also positive on average, but smaller. As expected, higher

incoming energy (LWd, Rn, G), evaporative demand (T and VPD), aerodynamical conductance (related to WS) and water supply (related to SM) generally favor higher NWL. In addition, there is a tendency to have less NWL (i.e. latent heat flux)

when sensible heat flux (SH) is higher, which is consistent with the partition of available energy. However, Spearman correlations at the majority of sites are smaller than 0.3. Reasons for this may include confounding effects among the analyzed drivers of NWL, observational uncertainty and a possible physiological control on nocturnal transpiration; e.g. the relationship of VPD with NWL might not increase monotonically if stomatal conductance decreases when VPD is high.

5    Although there is no clear dependency of the correlations on land cover, we note that croplands (some of them irrigated) often exhibit higher correlations with VPD and WS, while higher correlations with SM and LWd often correspond to short vegetation types. When analyzing data from summer months only, we find that correlations with VPD increase at forest sites, in particular at DBF. Also, the four sites with the highest correlations with SM are located in southern Arizona, an arid zone.

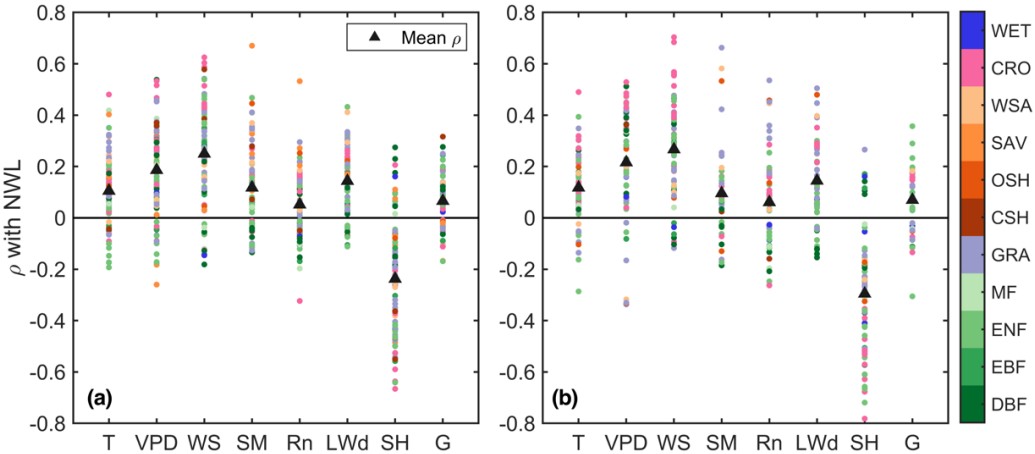

**Figure 4.** Spearman correlation ($\rho$) of 30-minute non-gap-filled nocturnal water loss (NWL) with air temperature (T), vapor pressure deficit (VPD), wind speed (WS), soil moisture (SM), net radiation (Rn), downward longwave radiation (LWd), sensible heat flux (SH) and ground heat flux (G) at FLUXNET sites. Panel (a) is for all data and (b) for summer months (JJA) at sites located above 30° N. Land cover types are deciduous broadleaf forest (DBF), evergreen needleleaf forest (ENF), evergreen broadleaf forest (EBF), mixed forest (MF), grassland (GRA), closed shrubland (CSH), open shrubland (OSH), savanna (SAV), woody savanna (WSA), cropland (CRO), and wetland (WET).

### 3.2 Climate model estimates of nocturnal water loss

The multi-model mean depicts an average $NWL_f$ of 7.9 % across all land grid cells excluding desert regions and Greenland (Fig. 5). The 5th percentile of the spatial distribution without deserts and Greenland is 1.8 %, and the 95th percentile is 13.2

20    %. In tropical regions $NWL_f$ is generally below the global average, even though NWL can e.g. surpass 80 mm yr$^{-1}$ in parts of the Amazon. Central and northern Europe, USA, China and India show similar regional averages of approximately 9 %. The models also suggest a high relevance of nocturnal water fluxes in Australia with an average $NWL_f$ of 13.1 %, and in the Mediterranean with 12 %. In most of Greenland and parts of Egypt the amount of dew or hoar frost is greater than the water lost through ET during the night. Interannual variability of $NWL_f$, given by the standard deviation of the 30-year time series

25    from the multi-model mean, is below 2 % on 95 % of land grid cells excluding deserts and Greenland. Finally, we focus in the northern midlatitudes (30–60° N) to analyze seasonality. The multi-model mean indicates that autumn (SON) is the

season with highest NWL on average (50.4 % of grid cells), whereas the lowest NWL typically corresponds to winter (DJF) (73 % of grid cells).

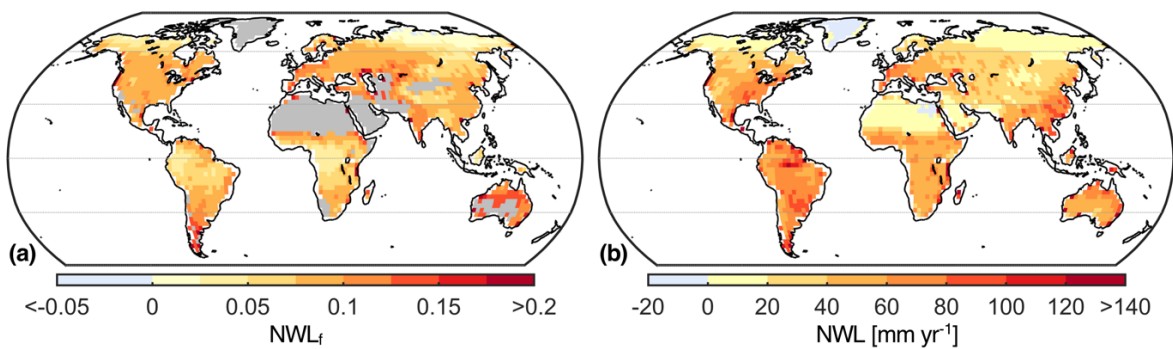

**Figure 5.** Map of multi-model mean $NWL_f$ (a) and NWL (b) on average over the period 1976–2005. Desert regions and Greenland are masked in (a) because of division by small numbers.

There are large discrepancies in $NWL_f$ between the different climate models (Fig. 6). The 95th percentile of the model ensemble is higher than 15 % in most of the globe, whereas the 5th percentile even shows negative values (i.e. dew is greater than nocturnal ET) in parts of the tropics and high latitudes. The central 90 % spread of the ensemble is almost everywhere larger than 10 %, and even greater than 20 % in southern South America, eastern Africa, India and Australia. This means that at certain locations some models simulate $NWL_f$ to be approximately zero, whereas estimates from other models are higher than 20 %. Even though the model differences in $NWL_f$ can originate from differences in total ET (e.g. in India), we also find differences in NWL generally ranging from 50 to 150 mm yr$^{-1}$ (see Fig. S1).

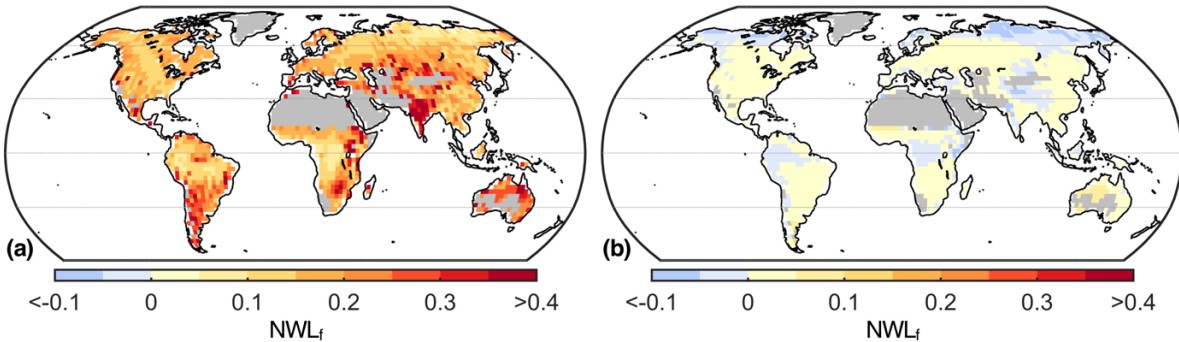

**Figure 6.** $NWL_f$ uncertainty within the climate model ensemble. (a) Map of the 95th percentile of the ensemble. (b) Map of the 5th percentile of the ensemble. Desert regions and Greenland are masked because of division by small numbers.

The complexity of CMIP5 models, together with the fact that not all models are equally well documented, hinders a straightforward assessment of potential factors contributing to the large inter-model differences in NWL. Nonetheless, we find a positive relation of climatological NWL and nighttime near-surface air temperatures across models (Fig. 7), indicating that models with high temperatures also tend to simulate high NWL. This correlation is present throughout the world and during the different seasons, although it decreases substantially in the Northern Hemisphere during summer (JJA).

Furthermore, we note that inmcm4, EC-EARTH, NorESM1-M and CNRM-CM5 are models with systematically low values of NWL throughout the globe; whereas GISS-E2-R, GISS-E2-H and MIROC5 tend to simulate the highest values of NWL (see Fig. S2).

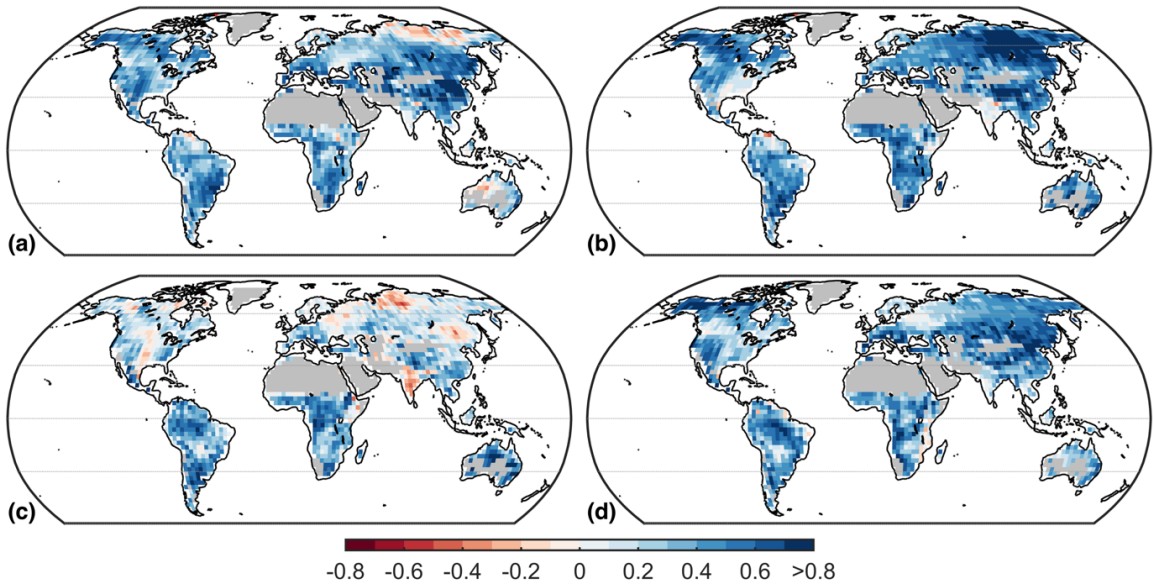

**Figure 7.** Pearson correlation at each grid cell between average NWL and nocturnal near-surface air temperature of climate models. Data corresponds to the period 1976–2005 from historical simulations. Correlations are computed separately for each season: (a) December–February, (b) March–May, (c) June–August, and (d) September–November. Desert regions and Greenland are masked for consistency.

Terrestrial $NWL_f$ is projected to increase towards the end of the century throughout the globe (Fig. 8). The average increase in the multi-model mean is 1.8 %, neglecting deserts and Greenland. Whereas NWL is projected to increase almost everywhere, this is not the case for total ET. The increase in $NWL_f$ in the Amazon, Central America, southern Africa and the Mediterranean is favored by a projected decrease in total ET. It is important to note that the spread of the model ensemble reduces confidence even in the sign of projected changes in NWL and total ET (Fig. S3). Lastly, we highlight the contribution of the nocturnal flux to projected changes in total ET. In more than half of all land grid cells, the projected change in NWL corresponds to 20 % or more of the absolute change in ET.

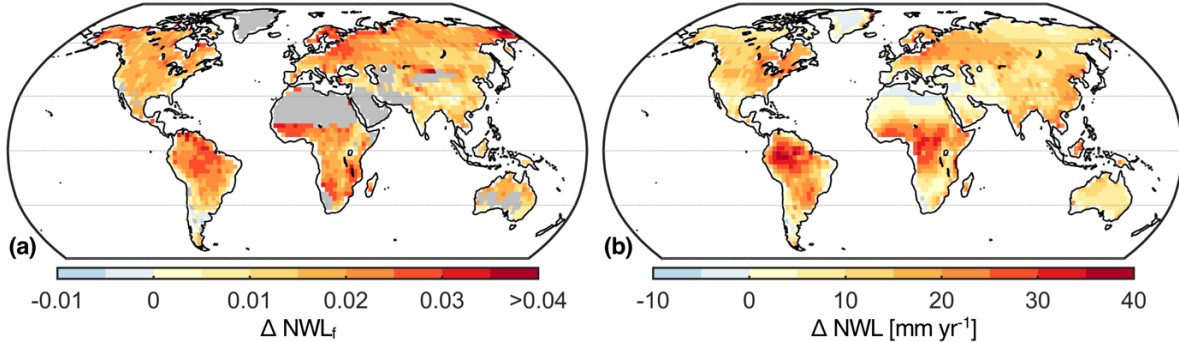

**Figure 8.** Multi-model mean of projected changes in $NWL_f$ (a) and NWL (b) for the period 2081–2100 relative to the period 1976–2005. Desert regions and Greenland are masked in (a) because of division by small numbers.

## 3.3 Comparison of observed and simulated nocturnal water loss

We compare the site-level EC observations to model estimates from the corresponding grid cells, despite the large difference in spatial resolution. Modelled $NWL_f$ generally shows an overestimation, although there are a few exceptions (Fig. 9a) – the average from the considered grid cells is 10.6 %, whereas the observational average is 7 %. Note once again the large discrepancies between individual models with an average spread of 20.5 % across locations calculated as the difference between the 97.5[th] percentile and 2.5[th] percentile. On the other hand, the estimated 95 % confidence interval of the EC observations is ± 0.15 % on average across sites. Interestingly, the multi-model mean has a smaller spread across sites than observations. This is partly explained by strong local discrepancies between individual models causing little variability in the multi-model mean; nonetheless, it could also be related to smoothing of cross-site differences in the much coarser spatial resolution of the models. At locations above 30° N, where most stations are found and seasonal differences are clearer, the simulated seasonal behavior agrees generally well with that of the EC data (Fig. 9b, see also Fig. S4). However, there is a noteworthy overestimation of the cases where the multi-model mean shows the lowest NWL to occur in summer, which is compensated by an underestimation for autumn and spring.

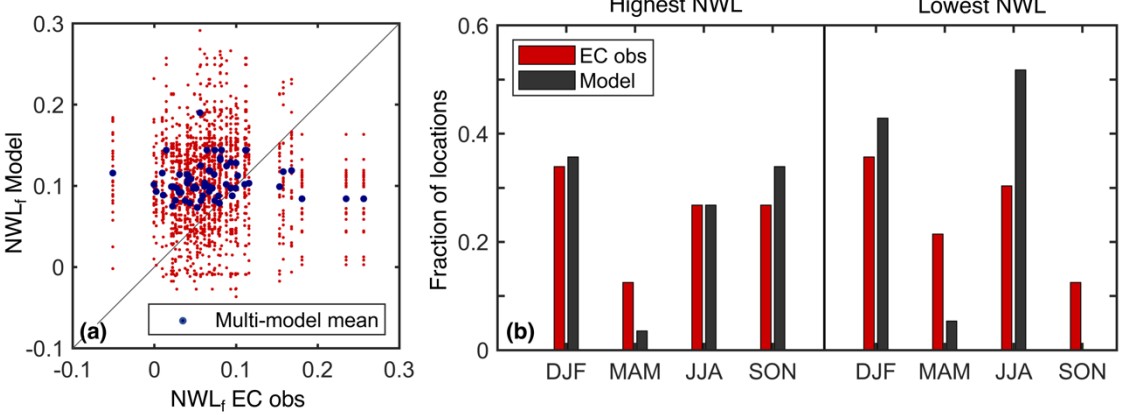

**Figure 9.** Comparison of observations with climate model simulations at the corresponding grid cells. (a) $NWL_f$ from EC observations versus model simulations at 64 locations. (b) Fraction out of 56 locations (i.e. FLUXNET sites or grid cells) above 30° N where each season has the highest or lowest NWL on average. Seasons are defined by the trimesters December–February (DJF), March–May (MAM), June–August (JJA) and September–November (SON).

## 4 Discussion and conclusions

Our average estimate of net nocturnal water loss relative to total evapotranspiration from 99 FLUXNET sites is 6.3 %. This is smaller than reported values around 10–25 % from published physiological studies (Zeppel et al., 2014). However, it is important to distinguish that our focus is on the net flux, i.e. evapotranspiration minus dew, whereas physiological studies refer only to transpiration. The results agree with the expectation of lower $NWL_f$ when dew is taken into account. In

addition, we recall that nocturnal measurements at FLUXNET stations can be affected by low-turbulence conditions, and therefore gap-filled and energy-balance-corrected data are used in the analysis. Future work could aim to disentangle the distinct fluxes of transpiration, evaporation from soil and canopy, sublimation and dew during the night.

We find that higher air temperature, vapor pressure deficit, wind speed, soil moisture and downward longwave radiation tend to favor higher NWL, although the correlations are rather low. Similar results were reported by Groh et al. (2019) at two sites in Germany. In addition, Dawson et al. (2007) found clear positive relationships between these conditions and nocturnal sap flow in woody plant species from different ecosystems; however, nocturnal sap flow could occur with no NWL, e.g., for capacitance refilling. Meanwhile, Zeppel et al. (2014) point to plant functional type, ecosystem type, and biotic temporal
characteristics like leaf or stand age, as possible additional factors influencing NWL. On the other hand, de Dios et al. (2015) found no temporal relation with vapor pressure deficit because of endogenous circadian regulation in an experiment with crops under controlled environmental conditions. Additionally, an increase in nocturnal sap flow and stomatal conductance was reported in two tree species under increased atmospheric $CO_2$ concentration, given sufficient soil moisture (Zeppel et al., 2011, 2012). Further research about the controls of NWL, and in particular nocturnal transpiration, is required.

The climate model ensemble has an average $NWL_f$ of 7.9 % over land, which is slightly higher than the observational estimate. Moreover, the overestimation is greater when considering only grid cells that contain FLUXNET sites. These relatively high multi-model mean estimates of $NWL_f$ are surprising given the literature that suggests models underestimate nocturnal stomatal conductance (e.g. Lombardozzi et al. 2017; Zeppel et al., 2014). Note that increasing model nocturnal
stomatal conductance would likely lead to even higher values of simulated $NWL_f$. Thus, it is possible that even if the mean simulated magnitude of nocturnal water loss is relatively accurate, the underlying processes may be misrepresented.

Our analysis indicates strong discrepancies between individual models in simulated $NWL_f$, which are much larger than the spatial and inter-annual variability. These discrepancies are related to differences in average nighttime temperature between
models. Simulations that disentangle nocturnal transpiration, evaporation (sublimation) from soil and canopy, and dew would be highly relevant to study the inter-model differences. Note that differences in NWL can represent a substantial fraction of model differences in total ET. Furthermore, these biases could affect boundary layer evolution and precipitation timing in models. Inter-model uncertainty also reduces confidence in the direction of change in NWL under global warming, despite the multi-model mean showing a projected increase throughout the world.

In conclusion, our study provides a comprehensive global overview of NWL – defined as nocturnal evapotranspiration minus dew formation – from observations and climate models. The magnitude of this flux suggests it can be important for the surface energy and water balances, and therefore relevant to consider in hydroclimate analyses. Future research about NWL focused at seasonal and shorter timescales could address its influence on climate impacts during extreme conditions

(e.g., Duarte et al., 2016; Groh et al., 2019). Finally, ongoing development and expansion in sensing water and energy fluxes are expected to help address the uncertainties we have highlighted around NWL through continued research on this topic.

*Data and code availability.* The FLUXNET2015 Tier 1 dataset is available at https://fluxnet.fluxdata.org/data/fluxnet2015-
5  dataset/. Table A1 indicates the specific sites considered for the analysis. The CMIP5 data used in this study are available at https://esgf-node.llnl.gov/projects/esgf-llnl/. Detailed inputs for the search query are as follows: Model (see Table A2), Experiment (historical, rcp85), Time Frequency (3hr), Ensemble (see Table A2), Variable (hfls, tas). Processed hourly data from the co-located lysimeter and EC tower at Rietholzbach, as well as accompanying meteorological data, and scripts used for the analysis are available at https://doi.org/10.3929/ethz-b-000370968.

10  **Appendix A: List of FLUXNET sites and climate models used in the analysis**

**Table A1.** FLUXNET sites from the FLUXNET2015 dataset employed for the analysis. Included sites provide energy balance corrected measurements of latent heat flux during at least three years. The SITE_ID is indicated here, whereas a full description of each site is available at https://fluxnet.fluxdata.org/sites/site-list-and-pages/. Additionally, the number of years of data and average energy balance corrected $NWL_f$ for each site is provided.

| SITE_ID | # of years | $NWL_f$ | SITE_ID | # of years | $NWL_f$ | SITE_ID | # of years | $NWL_f$ |
|---------|-----------|---------|---------|-----------|---------|---------|-----------|---------|
| AR-SLu | 3 | 0.158 | CN-HaM | 3 | 0.026 | IT-Tor | 7 | 0.051 |
| AT-Neu | 11 | 0.023 | CZ-wet | 9 | 0.040 | NL-Hor | 8 | 0.074 |
| AU-ASM | 4 | 0.081 | DE-Geb | 14 | 0.010 | NL-Loo | 18 | 0.082 |
| AU-Ade | 3 | 0.042 | DE-Gri | 11 | 0.031 | RU-Fyo | 17 | 0.011 |
| AU-Cpr | 5 | 0.057 | DE-Hai | 13 | -0.051 | SD-Dem | 5 | 0.100 |
| AU-Cum | 3 | 0.067 | DE-Kli | 11 | 0.041 | SN-Dhr | 4 | 0.103 |
| AU-DaP | 7 | 0.023 | DE-Lkb | 5 | 0.116 | US-AR1 | 4 | 0.111 |
| AU-DaS | 7 | 0.053 | DE-Obe | 7 | 0.040 | US-AR2 | 4 | 0.088 |
| AU-Dry | 7 | 0.061 | DE-RuR | 4 | 0.064 | US-ARM | 10 | 0.067 |
| AU-Emr | 3 | 0.081 | DE-RuS | 4 | 0.112 | US-Blo | 11 | 0.023 |
| AU-Fog | 3 | 0.122 | DE-Seh | 4 | 0.112 | US-Cop | 7 | 0.044 |
| AU-Gin | 4 | 0.041 | DE-SfN | 3 | 0.045 | US-GBT | 8 | 0.256 |
| AU-How | 14 | 0.035 | DE-Tha | 19 | 0.073 | US-GLE | 11 | 0.235 |
| AU-RDF | 3 | 0.049 | DK-Sor | 19 | -0.023 | US-KS2 | 4 | 0.034 |
| AU-Rig | 4 | 0.067 | ES-LgS | 3 | 0.105 | US-Los | 15 | 0.034 |
| AU-Stp | 7 | 0.061 | FI-Hyy | 19 | 0.036 | US-MMS | 16 | 0.002 |
| AU-Tum | 14 | 0.056 | FI-Jok | 4 | 0.021 | US-Me2 | 13 | 0.102 |
| AU-Wac | 4 | 0.168 | FR-Gri | 10 | 0.093 | US-NR1 | 17 | 0.181 |
| AU-Whr | 4 | 0.068 | FR-LBr | 13 | 0.043 | US-Ne1 | 13 | 0.032 |

| | | | | | | | | |
|---|---|---|---|---|---|---|---|---|
| AU-Wom | 3 | 0.088 | FR-Pue | 15 | 0.050 | US-Ne2 | 13 | 0.030 |
| AU-Ync | 3 | 0.041 | IT-BCi | 11 | 0.133 | US-Ne3 | 13 | 0.033 |
| BE-Bra | 19 | 0.027 | IT-CA2 | 4 | 0.044 | US-Prr | 4 | 0.058 |
| BE-Lon | 11 | 0.027 | IT-CA3 | 4 | 0.027 | US-SRG | 7 | 0.095 |
| BE-Vie | 19 | 0.015 | IT-Col | 19 | 0.045 | US-SRM | 11 | 0.078 |
| BR-Sa3 | 5 | 0.022 | IT-Cp2 | 3 | 0.020 | US-Syv | 14 | 0.045 |
| CA-Qfo | 8 | 0.047 | IT-Cpz | 13 | 0.031 | US-Ton | 14 | 0.039 |
| CA-SF1 | 4 | 0.057 | IT-Lav | 12 | 0.153 | US-Twt | 6 | 0.123 |
| CA-SF2 | 5 | 0.074 | IT-MBo | 11 | 0.021 | US-Var | 15 | 0.030 |
| CA-SF3 | 6 | 0.038 | IT-Noe | 11 | 0.147 | US-WCr | 16 | -0.001 |
| CH-Cha | 10 | 0.071 | IT-PT1 | 3 | 0.027 | US-Whs | 8 | 0.059 |
| CH-Dav | 18 | 0.099 | IT-Ren | 16 | 0.049 | US-Wkg | 11 | 0.067 |
| CH-Fru | 10 | 0.093 | IT-Ro2 | 11 | 0.017 | ZA-Kru | 11 | 0.032 |
| CN-Cng | 4 | 0.080 | IT-SRo | 14 | 0.058 | ZM-Mon | 10 | 0.053 |

**Table A2.** Climate models or model configurations employed for the analysis. Note that there are slightly variations depending on time period / scenario and on variable under consideration.

| Model | Simulation | 1976–2005: Historical | | 2081–2100: RCP8.5 |
|---|---|---|---|---|
| | | Latent heat flux | Temperature | Latent heat flux |
| ACCESS1-0 | r1i1p1 | X | X | X |
| ACCESS1-3 | r1i1p1 | X | X | X |
| bcc-csm1-1 | r1i1p1 | X | X | X |
| bcc-csm1-1-m | r1i1p1 | X | X | X |
| BNU-ESM | r1i1p1 | X | X | X |
| CCSM4 | r6i1p1 | X | X | X |
| CMCC-CM | r1i1p1 | X | X | X |
| CNRM-CM5 | r1i1p1 | X | X | X |
| EC-EARTH | r2i1p1 | X | X | X |
| FGOALS-g2 | r1i1p1 | X | X | X |
| FGOALS-s2 | r1i1p1 | X | | |
| GFDL-CM3 | r1i1p1 | X | X | X |
| GFDL-ESM2G | r1i1p1 | X | X | X |
| GFDL-ESM2M | r1i1p1 | X | X | |
| GISS-E2-H | r6i1p1 | X | X | X |
| GISS-E2-R | r6i1p1 | X | X | X |
| HadGEM2-ES | r2i1p1 | X | X | |
| inmcm4 | r1i1p1 | X | X | X |

| | | | | |
|---|---|---|---|---|
| IPSL-CM5A-LR | r1i1p1 | X | X | X |
| IPSL-CM5A-MR | r1i1p1 | X | X | X |
| MIROC-ESM | r1i1p1 | X | X | X |
| MIROC-ESM-CHEM | r1i1p1 | X | X | X |
| MIROC5 | r1i1p1 | X | X | X |
| MRI-CGCM3 | r1i1p1 | X | X | X |
| MRI-ESM1 | r1i1p1 | X | X | |
| NorESM1-M | r1i1p1 | X | X | X |

*Author contributions.* RSP, LG and SIS conceived the idea and designed the study. SIS acquired the funding to carry out the research. DM collected and processed the co-located lysimeter and EC data from the Swiss site. RSP processed the FLUXNET2015 and CMIP5 data. RSP performed the analysis and wrote the manuscript with contributions from all authors throughout the study. All authors discussed the results, read and reviewed the manuscript.

*Competing interests.* The authors declare that they have no conflict of interest.

**Acknowledgements**

We acknowledge partial support from the H2020 CRESCENDO project (grant agreement 641816), and from the European Research Council (ERC) DROUGHT-HEAT project funded by the European Community's Seventh Framework Programme (grant agreement FP7-IDEAS-ERC-617518). This work used eddy covariance data acquired and shared by the FLUXNET community, including these networks: AmeriFlux, AfriFlux, AsiaFlux, CarboAfrica, CarboEuropeIP, CarboItaly, CarboMont, ChinaFlux, Fluxnet-Canada, GreenGrass, ICOS, KoFlux, LBA, NECC, OzFlux-TERN, TCOS-Siberia, and USCCC. The ERA-Interim reanalysis data are provided by ECMWF and processed by LSCE. The FLUXNET eddy covariance data processing and harmonization was carried out by the European Fluxes Database Cluster, AmeriFlux Management Project, and Fluxdata project of FLUXNET, with the support of CDIAC and ICOS Ecosystem Thematic Center, and the OzFlux, ChinaFlux and AsiaFlux offices. We appreciate the substantial and exhaustive work carried out to provide best estimates of the fluxes in the FLUXNET2015 dataset. We acknowledge the World Climate Research Program's Working Group on Coupled Modelling, which is responsible for the Coupled Model Intercomparison Project (CMIP), and we thank the climate modelling groups for producing and making available their model output. For CMIP, the US Department of Energy's Program for Climate Model Diagnosis and Intercomparison provides coordinating support and led development of software infrastructure in partnership with the Global Organization for Earth System Science Portals. We thank Urs Beyerle for downloading the CMIP5 data.

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
