# Peer review of "Terrestrial Water Loss at Night: Global Relevance from Observations and Climate Models"

_Hydrology and Earth System Sciences, 2019_

## Referee Comment (RC1) · Anonymous Referee #1 · 29 Jul 2019

The overall focus of this paper is interesting; nocturnal evapotranspiration is an under-appreciated part of the hydrologic cycle that represents water loss without accompanying carbon gain (something that many resource managers might like to avoid). Thus, the result showing that nocturnal water loss (or NWL) represents a significant fraction of total ET across a wide range of biomes is likely of interest to a wide audience. The comparison of observed and modeled NWL rates is interesting in that, while the total magnitude of NWL is relatively similar between data and models (6.3 versus 7.9%), the relationship between modeled and observed NWL rates is virtually non-existing across sites (e.g. Fig 8a). This suggests some process-level room for improvement in the models.

Overall, I found that the study was largely exploratory; the mechanistic explanations

were limited to a simple spearman correlation analysis (e.g. Fig 4) of observations, and little discussion of how mechanistic representation of key processes in the models might affect the inter-model variability. While purely objective-oriented exploration of network level data can be useful, at the same time, better closing the gap between observations and models requires that underlying mechanisms be understood and carefully linked.

Towards that end, I have a few suggestions below for enhancing the mechanistic perspective of the paper that could ultimately leave the reader with a better understanding of not only how much water is lost at night, but also why this happens at different rates across ecosystems and models.

1. Much of the introduction reads like a list of previously published papers on the topic. While it is important to acknowledge this prior work, it would also be quite helpful to review for the reader the various mechanisms that could contribute to high NWL (e.g. not only incomplete stomatal closure, but also non-negligible cuticular conductance, and nocturnal evaporation from soils and canopies, snow sublimation). From there, it may even be possible to craft some expectations about in which ecosystems, and when, NWL should be especially prominent in the observations.

2. Likewise, some discussion about how the different models treat relevant processes and parameters could allow for a more informed understanding of why they differ so widely in their estimation of NWL. The authors suggest that most of the models employ the Ball-Berry stomatal conductance model (e.g. Page 2 Line 23)... Is this true for the models studied here, and do they adopt similar formulations for the intercept of this model? Knowing precisely how these models treat nocturnal conductance would go a long way towards understanding if the cross-model differences are linked to model ecophysiological representation.

3. Related to (2), I found it quite interesting that model differences were related to near surface temperature (page 12, line 6); unfortunately, this result is buried in the

SI. I would urge the authors to bring this result into the main text, and also expand the discussion about why this correlation exists.

4. The mechanistic analysis of the data is limited to correlations between NWL rates and VPD, wind speed, and soil moisture. I agree that these are important drivers of ET. However, even though incident solar radiation is zero at night, energy is still required to drive ET at night. The paper would strongly benefit from a discussion of where this energy comes from, which would require consideration of sensible and ground heat fluxes. . .and thus provide additional mechanistic insight.

I also had a few concerns about the treatment of the flux data.

1. The analysis relies on datasets that are largely gapfilled. While gapfilled data are necessary for estimating annual sums, they are not required for exploring relationships between ET observations and meteorological drivers. Can the authors repeat the analysis for Figure 4, but using only data that pass the quality control test?

2. The flux observations have been corrected so that the energy budget is fully closed. This correction is quite controversial in the flux community, especially since the mechanisms causing the lack of energy balance closure are still not fully known (and at least one school of thought suggests that much of the problem could be linked to sensible heat flux). Thus, I urge the authors to repeat the analysis without the energy balance correction, and include a summary of those results (at least in the SI).

A few other comments:

Page 1, Lines 15-20. Much of the first paragraph is not well written. It states that ET is an important process, but does not tell us specifically why we should be concerned about NWL specifically. Moreover, the logic is not clear: the authors tell us that VPD, temperature and wind speed affect ET, and that half of the diurnal cycle is night, therefore NWL can be important. This conclusion does not follow from the premise (missing is a discussion about the prevalence of VPD, temperature and wind speed conditions

that could generate substantial nocturnal ET).

Page 3, Lines 1-5: This paragraph, which discusses the overall objective of the study, is quite short and lacks detail. Here would be an excellent place to discuss some expectations as to how NWL relates to "different meteorological and land cover conditions." The model-data comparison should also be mentioned here, and perhaps expectations offered as to which models are best equipped to accurately describe NWL patterns.

Section 2.1.2: Are the Fluxnet2015 data corrected for LE storage terms at night? Is this important?

Page 7, Line 4: The relationship between VPD and NWL may not be linear if stomatal conductance decreases when VPD is high, even at night.

Page 11, lines 20: The discussion of nocturnal stomatal conductance here is interesting; it strikes me as a bit of a missed opportunity not to explore patterns of nocturnal surface conductance from the data (it is relatively straightforward to invert flux tower ET measurements with the Penman-Montieth equation to obtain half-hourly surface conductance, e.g. see Wever et al. 2002 https://doi.org/10.1016/S0168-1923(02)00041-2). Doing so would illuminate whether cross-site differences in NWL are driven largely by biotic versus abiotic factors.

Figure 7: Considering that the models and data don't agree at all on the site level, can we really have much confidence in these future projections?

---

## Referee Comment (RC2) · Anonymous Referee #2 · 13 Aug 2019

Padrón and others analyze nocturnal evapotranspiration measurements from eddy co-variance and estimates from models. The analysis is interesting and certainly novel although a few methodological points need to be reconsidered in my opinion, and the text could be improved in multiple instances. Sentences like 'Lombardozzi et al. (2017) compiled evidence of this from 204 species' aren't particularly instructive. What did they find? In the paragraph at the bottom of page 1 try to make the scientific findings, not the authors, the subject of the sentences. For a discussion of this see https://schimelwritingscience.wordpress.com and the associated book. A more powerful way to synthesize the literature, which would make the present manuscript more citeable, would be to synthesize existing studies in a table to help further motivate the present analysis and be more comprehensive. The points about dew and hoar frost are

Printer-friendly version

Discussion paper

[Figure]

great. P 2 line 22: disentangle aerodynamic vs. surface conductances more clearly. The surface has both stomatal and boundary-layer resistances. 2.1.1: Why is the 10 W m-2 threshold used to differentiate between day and night? Sensors have uncertainty but the solar zenith angle can be calculated with extreme accuracy for environmental science applications. Are results sensitive to the 10 W m-2 threshold? I see that a zenith angle-based analysis is done in section 2.1.2 (sun up and sun down). Why are different approaches used? What are the 'cases described by Hirschi et al. (2017)'? P 3 line 30: using a static value for the latent heat of vaporization is fine, but it's easy to add its temperature sensitivity to add a bit more accuracy in the latent heat to water flux conversion. 2.1.2: The Bowen-ratio-based assumption is a bit problematic; there is extensive evidence that undermeasured sensible heat flux from large eddies plays a large role in lack of energy balance closure. That being said, these factors are less important at night where low-level jets and decoupling of the eddy covariance sensors and the canopy often dominate. 2.1.2: instead of emphasizing caution, perhaps don't use gapfilled fluxes for the analysis. This is a hard thing to do at night when eddy covariance data are often less reliable than many people believe. Thinking broadly, is 'nocturnal water flux' better than NWL given that water can be both lost and gained (but is admittedly a net loss over the time scales mostly investigated here). 3.1: why is the second threshold chosen? Is it appropriate for the site or just simply half of the previous threshold? Fig. 2 and elsewhere: what are representative uncertainties of the site-level NWL measurements/estimates? This statement should be in the Methods: "These annual mean values are computed from monthly climatologies obtained by omitting months with half or more of missing latent heat flux data." In general, the assumptions made in the flux processing for NWL for the FLUXNET2015 database needs to be explained in more detail. The statement 'Nonetheless, deciduous broadleaf forests (DBF) have an overall lower NWLf, whereas evergreen needleleaf forests (ENF) include most cases with higher NWLf' suggests to me that perhaps difficulties in measuring the surface-atmosphere flux is partly responsible here. ENF needles are more closely coupled to the atmosphere on account of their smaller dimensions and I can't think of a discernable reason why DBF would have particularly low NWL. Although perhaps relative NWL given that they are frequently found in mesic regions. Figure 4 is tricky to look at. I'm curious to know if there is a more logical way of presenting these complex data. The analysis of models is interesting and the degree of discrepancy is surprising.

---

## Referee Comment (RC3) · Anonymous Referee #3 · 18 Aug 2019

The paper is handling an important topic in ecohydrology – the nocturnal water loss of ecosystems. The tools used in comparison are appropriate but not convincingly comprehensive. The processes that might cause differences in derived NWL between EC measurements and modelled data should be investigated more thoroughly by diving e.g. into variable footprints, processes handled in the models, gap-filling problems for ET from EC during night, and general night-time problems present in EC data. I clearly would desire uncertainty estimates for NWL especially as we a dealing with very low fluxes. Fortunately, NWL can only take place under well mixed conditions which gives trust in the nocturnal EC data used for the analysis. But we have to consider that ET (measurements and post-processing) has unfortunately hardly been the main focus of the FLUXNET data set. So we should be aware that so far we do not have well established gap filling procedures for ET at night, especially under stable conditions. Thus, the paper lacks uncertainty estimates for the nocturnal fluxes determined by EC.

Specific comments: Introduction: What are the processes causing nocturnal water loss? Which kind of energy is converted into ET at night? And why is it so important to deal with? It should be mentioned that a water loss is accepted (during day-time) by gaining carbon. Is there any advantage for the plants or the ecosystem to loose water at night? Or just no possibility to avoid? The authors mainly summarize previous work here. Page 2, Line 18/19: 'Both ET and dew correspond to a latent heat flux and can prove difficult to disentangle depending on the temporal resolution of the data.' These fluxes are in opposite direction, even if the net ET might comprise a combination of both, for energetic reasons these processes hardly occur simultaneously. Could you describe more clearly what exactly is meant?

Page 3, Line 21: if weight increase without rain measured is considered as rain or snow, we have to ask how reliable are the rain measurements? Or otherwise you should provide any further explanation for the procedure. And maybe the frequency of occurrence or the amount of water switched from dew to rain.

Page 3, Line 27ff: we have to consider that ET has never been the main focus of the FLUXNET data set. This statement should not imply that all ET data from FLUXNET are less reliable. But we should be aware that so far we do not have well established gap filling for ET at night, especially under stable conditions. Fortunately, NWL can only take place under well mixed conditions which gives trust in the nocturnal EC data used for the analysis. Most probably the majority of the data used for the analysis were measured anyway. But it would be quite interesting to see the relation of measured and gap-filled data used for the data-analysis, not only for the Rietholzbach site but also for the FLUXNET analysis. This information gives also a hint related to the uncertainty of the derived nocturnal fluxes.

Page 4, Line 8: for night-time data? Page 4, Line 14-15: move this sentence to the

acknowledgements, even though appreciated by myself

Page 5, Lines 4ff: this section should be improved by quantitative uncertainty values. Page 6, Line 14: '…across sites cannot easily be explained by annual average….'

Page 7, Line 1: can you be sure that EC data are reliable under 'snowy and windy conditions'? EC assumption might not be fulfilled, sonic data are often disturbed under such conditions.

Page 10, lines 1ff: for EC estimates no uncertainty is considered. How large are the uncertainties related to the fluxes under consideration? Page 11, lines 5ff: here it is correctly said that nocturnal measurements can be affected by low-turb conditions. But nocturnal fluxes are not treated by the energy-balance correction, as also correctly said before. In the discussion part, also the uncertainty of EC data should be discussed.

Figure 1, caption: should include the site name Figure 2: caption to be extended. What exactly is show? Always consider that reader often concentrate on the figures of a paper only and thus need more information. In addition, in c), the colours of the tiny dots are difficult to distinguish with normal page size. But I also fear, this is not a 'spatial distribution' but rather a 'distribution of sites with ….'

---

## Author Response (AR1)

**Response to editor**

*Dear authors,*
*Thank you for carefully responding to all referee comments and for your proposed improvements to the manuscript. All referees see a great value in your work but also raised numerous points to address. I believe that if the referees comments are addressed adequately, as indicated in most of your responses, the paper will be indeed a valuable puzzle piece for understanding surface-atmosphere water exchange. In addition to the changes you proposed, could you consider the following?*

*Dear Stan, we appreciate your feedback.*

*1) Referee 1 critices that the mechanisms resulting in modelled NWL were not explained, while reference is made to an alleged under-estimation of nocturnal stomatal conductance in the models without going into details. Even though a detailed analysis of how NWL is simulated in the different models may be out of scope for this paper, it would be helpful if you provided NWL results for each model in addition to the multi-model mean values, or at least a ranking of the models, so that the readers could verify for themselves what might cause the spread in the simulations. The CMIP5 data does actually distinguish between water_evaporation_flux_from_soil, water_evaporation_flux_from_canopy and transpiration_flux, so you could at least verify what proportion of simulated NWL comes from transpiration as opposed to evaporation in the models. I think that this would be within the scope of the study, which is to provide a "general overview of NWL across the globe from observations and climate models".*

*We expanded the discussion about model discrepancies in the text indicating which models tend to have systematically high and low values of NWL, and also added Fig. S2 showing the ranking of all analyzed models.*

*We agree that it would be relevant and interesting to disentangle the different fluxes contributing to NWL in the models, however, these data are not available in the CMIP5 archive with a 3-hour temporal resolution. Therefore, we are not able to compute the contribution of the individual fluxes during the night.*

*2) The explanation of the process underlying NWL should include an energy balance consideration. You mention the influence of air temperature, VPD and wind speed on NWL, but what about sensible heat flux, soil heat flux (also soil temperature), and longwave radiation?*

*We modified Fig. 4 and expanded the corresponding text to also analyze correlations of NWL with the suggested variables.*

*3) Please make sure that all your responses to the referees are also reflected in the manuscript, as the referees' comments likely reflect your future readers' thoughts. For example, your response to Referee #1 that the data does not account for LE storage might also be an important information for the reader.*

*We appreciate the suggestion and modified the text accordingly.*

*4) I agree with Referee #1 that Figure 7 does not add much value to the paper and could be*

*removed, unless you would like to make a strong point that the models do not only disagree about the magnitude of NWLf but also its trend.*

*We removed two panels from Fig. 7 and modified the text. We convey two main points about future projections of NWL: (i) NWL is projected to increase everywhere with an average of 1.8 %, although with a substantial inter-model spread. (ii) Changes in NWL contribute substantially to projected changes in total ET.*

*5) In your response to Referee #3, you propose that possible ecological advantages of NWL include capacitance refilling, embolism removal and hydraulic redistribution, among others. Hydraulic redistribution, capacitance refilling and embolism removal may explain nocturnal sap flow, but not water loss. The difference between sap flow and NWL sensed by EC towers should be emphasized more prominently throughout the text.*

*We now clarify that these possible advantages are for nocturnal sap flow, and not necessarily NWL. We modified the text to emphasize the difference between nocturnal sap flow and NWL.*

*6) Your response to Referee #3 about P7L1: It should be easy to verify if these sites include more gap-filled data than the average, and to mention this in the text.*

*We now include this information in the text.*

*7) Data availability and reproducibility of results: Thank you for providing the original lysimeter data. However, for your analysis to become reproducible, it would be important to also provide the scripts that were used to analyze both the Fluxnet and the CMIP5 data. In addition, when I opened the link given for the CMIP5 data, I had to click on CMIP5 in the list of data, create an account and eventually landed on a search page for data. Could you please provide accurate instructions on how to access the exact data you used for the study? Is there a specific search query that would take the reader to the right data? Same for the Fluxnet data: which 99 stations did you use, and which years of each station?*

*We updated our data availability statement to clarify these points about the specific FLUXNET and CMIP5 data used in the study. Reproducibility of the analysis should be feasible based on the information provided in the manuscript. No software or model code was developed. We do not consider relevant to provide several customized scripts for data selection and manipulation, e.g. computing climatologies, plotting, or computing correlations.*

**List of main changes to the manuscript**

- The introduction was rewritten, including the addition of Table 1 summarizing results from previous studies.
- Addition of uncertainty estimates for the FLUXNET observations of NWL: Changes to text and Fig. 2.
- Addition of temperature, radiation, sensible and ground heat flux to the analysis of factors influencing NWL in Fig. 4.
- Addition of one paragraph about model discrepancies in NWL and Fig. 7 (previously Fig. S2) showing the relation between model differences in NWL and model differences in nocturnal temperature.

**Response to reviewers**

*The initial point-by-point response to the reviewers during the interactive discussion is included below.*

**Reviewer 1:**

*The overall focus of this paper is interesting; nocturnal evapotranspiration is an under-appreciated part of the hydrologic cycle that represents water loss without accompanying carbon gain (something that many resource managers might like to avoid). Thus, the result showing that nocturnal water loss (or NWL) represents a significant fraction of total ET across a wide range of biomes is likely of interest to a wide audience. The comparison of observed and modeled NWL rates is interesting in that, while the total magnitude of NWL is relatively similar between data and models (6.3 versus 7.9%), the relationship between modeled and observed NWL rates is virtually non-existing across sites (e.g. Fig 8a). This suggests some process-level room for improvement in the models.*

*We appreciate the positive opinion about the relevance of our manuscript.*

*Overall, I found that the study was largely exploratory; the mechanistic explanations were limited to a simple spearman correlation analysis (e.g. Fig 4) of observations, and little discussion of how mechanistic representation of key processes in the models might affect the inter-model variability. While purely objective-oriented exploration of network level data can be useful, at the same time, better closing the gap between observations and models requires that underlying mechanisms be understood and carefully linked.*

*Towards that end, I have a few suggestions below for enhancing the mechanistic perspective of the paper that could ultimately leave the reader with a better understanding of not only how much water is lost at night, but also why this happens at different rates across ecosystems and models.*

*1. Much of the introduction reads like a list of previously published papers on the topic. While it is important to acknowledge this prior work, it would also be quite helpful to review for the reader the various mechanisms that could contribute to high NWL (e.g. not only incomplete stomatal closure, but also non-negligible cuticular conductance, and nocturnal evaporation from soils and canopies, snow sublimation). From there, it may even be possible to craft some expectations about in which ecosystems, and when, NWL should be especially prominent in the observations.*

*The introduction is modified and extended according to the suggestion.*

*2. Likewise, some discussion about how the different models treat relevant processes and parameters could allow for a more informed understanding of why they differ so widely in their estimation of NWL. The authors suggest that most of the models employ the Ball-Berry stomatal conductance model (e.g. Page 2 Line 23). . . Is this true for the models studied here, and do they adopt similar formulations for the intercept of this model? Knowing precisely how these models treat nocturnal conductance would go a long way towards understanding if the cross-model differences are linked to model ecophysiological representation.*

*We completely agree. We expanded the discussion on factors affecting inter-model variability and introduced a new figure (previously Fig. S2 in the Supporting Information). Yes, we note that most of the analyzed climate models' stomatal conductance formulations are based on the Ball-Berry model. Note that the complexity of CMIP5 models, and the fact that not all models are equally well documented, hinders a simple explanation of inter-model variability. In addition to how individual models represent nocturnal conductance, other factors such as planetary boundary evolution and soil parameterizations might also influence the inter-model variability. Thus, we consider this more detailed analysis to be outside the scope of our study, but nonetheless an interesting topic for a follow-up article.*

*3. Related to (2), I found it quite interesting that model differences were related to near surface temperature (page 12, line 6); unfortunately, this result is buried in the SI. I would urge the authors to bring this result into the main text, and also expand the discussion about why this correlation exists.*

*We appreciate the suggestion. We now include this as Figure 7 in the revised manuscript and expanded the discussion.*

*4. The mechanistic analysis of the data is limited to correlations between NWL rates and VPD, wind speed, and soil moisture. I agree that these are important drivers of ET. However, even though incident solar radiation is zero at night, energy is still required to drive ET at night. The paper would strongly benefit from a discussion of where this energy comes from, which would require consideration of sensible and ground heat fluxes. . . and thus provide additional mechanistic insight.*

*We appreciate the suggestion. We now include in Fig. 4 also the relation of NWL with net radiation, downward longwave radiation, sensible and ground heat flux. We additionally expanded the discussion accordingly.*

*I also had a few concerns about the treatment of the flux data.*

*1. The analysis relies on datasets that are largely gapfilled. While gapfilled data are necessary for estimating annual sums, they are not required for exploring relationships between ET observations and meteorological drivers. Can the authors repeat the analysis for Figure 4, but using only data that pass the quality control test?*

*This was already the case for Figure 4. We now clarify this in the text and figure caption.*

*2. The flux observations have been corrected so that the energy budget is fully closed. This correction is quite controversial in the flux community, especially since the mechanisms causing the lack of energy balance closure are still not fully known (and at least one school of thought suggests that much of the problem could be linked to sensible heat flux). Thus, I urge the authors to repeat the analysis without the energy balance correction, and include a summary of those results (at least in the SI).*

*We appreciate the insights. We now include this also in the manuscript and provide more information on the uncertainty of the EC fluxes. See modified version of Fig. 2 and corresponding changes to the text.*

*A few other comments:*

*Page 1, Lines 15-20. Much of the first paragraph is not well written. It states that ET is an important process but does not tell us specifically why we should be concerned about NWL specifically. Moreover, the logic is not clear: the authors tell us that VPD, temperature and wind speed affect ET, and that half of the diurnal cycle is night, therefore NWL can be important. This conclusion does not follow from the premise (missing is a discussion about the prevalence of VPD, temperature and wind speed conditions that could generate substantial nocturnal ET).*

*We reformulated the paragraph.*

*Page 3, Lines 1-5: This paragraph, which discusses the overall objective of the study, is quite short and lacks detail. Here would be an excellent place to discuss some expectations as to how NWL relates to "different meteorological and land cover conditions." The model-data comparison should also be mentioned here, and perhaps expectations offered as to which models are best equipped to accurately describe NWL patterns.*

*We reformulated and expanded the paragraph. In addition, note that our study follows an exploratory approach rather than specific hypothesis testing, which is why we do not provide any assumptions besides the known influence of abiotic factors like temperature, VPD and wind speed on evaporation/sublimation from the soil or canopy.*

*Section 2.1.2: Are the Fluxnet2015 data corrected for LE storage terms at night? Is this important?*

*The relevant data processing is described in the text and the referenced FLUXNET website. To our knowledge the FLUXNET2015 data does not account for LE storage in the air between the ground and measurement level.*

*Page 7, Line 4: The relationship between VPD and NWL may not be linear if stomatal conductance decreases when VPD is high, even at night.*

*We now also explicitly mention this in the text.*

*Page 11, lines 20: The discussion of nocturnal stomatal conductance here is interesting; it strikes me as a bit of a missed opportunity not to explore patterns of nocturnal surface conductance from the data (it is relatively straightforward to invert flux tower ET measurements with the Penman-Monteith equation to obtain half-hourly surface conductance, e.g. see Wever et al. 2002 https://doi.org/10.1016/S0168-1923(02)00041- 2). Doing so would illuminate whether cross-site differences in NWL are driven largely by biotic versus abiotic factors.*

*We find this suggestion very interesting and an excellent idea for a more specific study on surface conductance. Our main goal here is to provide a first more general overview of NWL across the globe from observations and climate models.*

*Figure 7: Considering that the models and data don't agree at all on the site level, can we really have much confidence in these future projections?*

*The inter-model variability of future $NWL_f$ projections is indeed large as shown in Fig. 7d and acknowledged on page 9 lines 10–12. Future studies could aim at reducing inter-model spread and constraining future projections.*

**Reviewer 2:**

*Padrón and others analyze nocturnal evapotranspiration measurements from eddy covariance and estimates from models. The analysis is interesting and certainly novel although a few methodological points need to be reconsidered in my opinion, and the text could be improved in multiple instances.*

*We appreciate the positive opinion about the relevance of our manuscript.*

*Sentences like 'Lombardozzi et al. (2017) compiled evidence of this from 204 species' aren't particularly instructive. What did they find? In the paragraph at the bottom of page 1 try to make the scientific findings, not the authors, the subject of the sentences. For a discussion of this see https://schimelwritingscience.wordpress.com and the associated book.*

*We appreciate the suggestion. We modified the text to improve the focus and readability.*

*A more powerful way to synthesize the literature, which would make the present manuscript more citable, would be to synthesize existing studies in a table to help further motivate the present analysis and be more comprehensive.*

*We appreciate the suggestion. We now introduce Table 1 in the revised manuscript to summarize nocturnal water loss estimates from the literature.*

*The points about dew and hoar frost are great.*

*Thank you.*

*P 2 line 22: disentangle aerodynamic vs. surface conductances more clearly. The surface has both stomatal and boundary-layer resistances.*

*We clarified this. We now provide a more complete description of the resistances included in models to compute latent heat flux.*

*2.1.1: Why is the 10 W m-2 threshold used to differentiate between day and night? Sensors have uncertainty but the solar zenith angle can be calculated with extreme accuracy for environmental science applications. Are results sensitive to the 10 W m-2 threshold? I see that a zenith angle-based analysis is done in section 2.1.2 (sun up and sun down). Why are different approaches used? What are the 'cases described by Hirschi et al. (2017)'?*

*Here we use this simple threshold because the focus is on the comparison of the lysimeter and EC data, and we wanted to be consistent with the comparison from Hirschi et al. (2017). The results are hardly sensitive to the 10 W $m^{-2}$ threshold.*

*We extended the sentence to clarify the meaning of 'cases described by Hirschi et al. (2017)'. It corresponds to cases when the tower is upwind of the sensor and thus disturbing the air flow.*

*P 3 line 30: using a static value for the latent heat of vaporization is fine, but it's easy to add its temperature sensitivity to add a bit more accuracy in the latent heat to water flux conversion.*

*Yes, we are aware of this, but for simplicity and to avoid dealing with possible missing temperature data, we assume a value of λ corresponding to a temperature of approximately 12 °C. We trade simplicity for a very small loss in accuracy. In addition, note that we do not incur in a highly biased error, given that temperatures are likely to be sometimes greater and sometimes less than 12 °C.*

*2.1.2: The Bowen-ratio-based assumption is a bit problematic; there is extensive evidence that undermeasured sensible heat flux from large eddies plays a large role in lack of energy balance closure. That being said, these factors are less important at night where low-level jets and decoupling of the eddy covariance sensors and the canopy often dominate.*

*We appreciate the insight. We now analyze the uncertainty of NWL estimates with and without the Bowen ratio assumption in Fig. 2.*

*2.1.2: instead of emphasizing caution, perhaps don't use gap-filled fluxes for the analysis. This is a hard thing to do at night when eddy covariance data are often less reliable than many people believe.*

*Gap-filled fluxes are required in order to obtain the total NWL estimates shown in Fig. 2. An alternative would be to estimate a mean hourly NWL rate from the non-gap-filled observations and obtain total sums by multiplying the mean by the total number of nighttime hours. However, this has its own disadvantages. Nonetheless results are rather similar with both options.*

*When analyzing the correlation of NWL with environmental conditions in Fig. 4 we do not employ gap-filled data.*

*Thinking broadly, is 'nocturnal water flux' better than NWL given that water can be both lost and gained (but is admittedly a net loss over the time scales mostly investigated here).*

*In a first draft we also used nocturnal water flux but decided that NWL is more appropriate to communicate our results.*

*3.1: why is the second threshold chosen? Is it appropriate for the site or just simply half of the previous threshold?*

*In this case is just half of the defined threshold value to provide an estimate of the sensitivity. We revised the text to make this clearer.*

*Fig. 2 and elsewhere: what are representative uncertainties of the site-level NWL measurements/estimates?*

*Figure 2 now also includes uncertainties of NWL estimates from FLUXNET sites. The text accompanying the analysis was modified to convey this point more clearly.*

This statement should be in the Methods: "These annual mean values are computed from monthly climatologies obtained by omitting months with half or more of missing latent heat flux data."

*Note that there is no specific "Methods" section in the structure of our manuscript. We thus think the location of the statement is appropriate.*

In general, the assumptions made in the flux processing for NWL for the FLUXNET2015 database needs to be explained in more detail.

*We expanded the text. Note also that a full description of the processing is provided at https://fluxnet.fluxdata.org/data/fluxnet2015-dataset/data-processing/, as indicated in the text.*

The statement 'Nonetheless, deciduous broadleaf forests (DBF) have an overall lower NWLf, whereas evergreen needleleaf forests (ENF) include most cases with higher NWLf' suggests to me that perhaps difficulties in measuring the surface-atmosphere flux is partly responsible here. ENF needles are more closely coupled to the atmosphere on account of their smaller dimensions and I can't think of a discernable reason why DBF would have particularly low NWL. Although perhaps relative NWL given that they are frequently found in mesic regions.

*We appreciate the insight and now include it in the text. Note that cross correlations and confounding factors might also be relevant.*

Figure 4 is tricky to look at. I'm curious to know if there is a more logical way of presenting these complex data.

*We increased the size of the symbol representing the mean to convey the main message. We expect that the text also helps to understand the Figure.*

The analysis of models is interesting, and the degree of discrepancy is surprising.

*Thank you.*

**Reviewer 3:**

The paper is handling an important topic in ecohydrology – the nocturnal water loss of ecosystems.

*We appreciate the positive opinion on the relevance of the manuscript.*

The tools used in comparison are appropriate but not convincingly comprehensive. The processes that might cause differences in derived NWL between EC measurements and modelled data should be investigated more thoroughly by diving e.g. into variable footprints, processes handled in the models, gap-filling problems for ET from EC during night, and general night-time problems present in EC data.

*We expanded our discussion of these points. Note that differences between EC and modelled data are expected due to the stark difference in spatial resolution. This was mentioned on page 10 lines 6–7, and now also in the introduction of the revised manuscript.*

*I clearly would desire uncertainty estimates for NWL especially as we are dealing with very low fluxes. Fortunately, NWL can only take place under well mixed conditions which gives trust in the nocturnal EC data used for the analysis. But we have to consider that ET (measurements and post-processing) has unfortunately hardly been the main focus of the FLUXNET data set. So, we should be aware that so far, we do not have well established gap filling procedures for ET at night, especially under stable conditions. Thus, the paper lacks uncertainty estimates for the nocturnal fluxes determined by EC.*

*We appreciate the insight. We acknowledge the difficulties to adequately measure latent heat flux during the night with EC systems, as mentioned on page 4, lines 12–16. The relatively good agreement of NWL climatology from EC and lysimeter data suggests that meaningful estimates can be obtained with EC measurements.*

*We now include some uncertainty estimates of EC NWL in Figure 2, based also on comments from the other reviewers.*

*Specific comments: Introduction: What are the processes causing nocturnal water loss? Which kind of energy is converted into ET at night? And why is it so important to deal with? It should be mentioned that a water loss is accepted (during day-time) by gaining carbon. Is there any advantage for the plants or the ecosystem to lose water at night? Or just no possibility to avoid? The authors mainly summarize previous work here.*

*We expanded the introduction to include the suggested points. It now includes the following statements: "Nocturnal water loss may occur as evaporation from soil and canopy, snow sublimation, or plant transpiration through stomatal and cuticular conductance. It is also recognized that vapor pressure deficit, temperature, wind speed, longwave radiation and surface resistance influence nocturnal ET (Monteith, 1965; Penman, 1948)" and "Possible advantages of nocturnal sap flow include capacitance refilling, embolism removal, nutrient uptake, hydraulic redistribution and oxygen supply (Zeppel et al., 2014), whereas it remains unclear if Tr with no associated carbon gain has any benefits for vegetation or is simply unavoidable".*

*Page 2, Line 18/19: 'Both ET and dew correspond to a latent heat flux and can prove difficult to disentangle depending on the temporal resolution of the data.' These fluxes are in opposite direction, even if the net ET might comprise a combination of both, for energetic reasons these processes hardly occur simultaneously. Could you describe more clearly what exactly is meant?*

*We reformulated the sentence to clarify this. We agree that it is likely that they do not occur simultaneously, but they are likely to co-occur during e.g. the 3-hour temporal resolution of the modelled data. Thus, for simplicity and conciseness, we focus on the net flux.*

*Page 3, Line 21: if weight increase without rain measured is considered as rain or snow, we have to ask how reliable are the rain measurements? Or otherwise you should provide any further explanation for the procedure. And maybe the frequency of occurrence or the amount of water switched from dew to rain.*

*We include one additional explanatory sentence. It is possible that because of the 0.1 mm resolution of the rain gauge, no precipitation is recorded, while the lysimeter mass increases.*

*Also, dew formation might be more favored to occur over vegetation than rain gauges. Moreover, it is also possible that the registered weight increase was due to something different than water input, e.g. a bird. In any case, the frequency is ~4 % of the hourly intervals when the dew was estimated, and the amount is ~4 mm yr$^{-1}$.*

Page 3, Line 27ff: we have to consider that ET has never been the main focus of the FLUXNET data set. This statement should not imply that all ET data from FLUXNET are less reliable. But we should be aware that so far, we do not have well established gap filling for ET at night, especially under stable conditions. Fortunately, NWL can only take place under well mixed conditions which gives trust in the nocturnal EC data used for the analysis. Most probably the majority of the data used for the analysis were measured anyway. But it would be quite interesting to see the relation of measured and gap-filled data used for the data-analysis, not only for the Rietholzbach site but also for the FLUXNET analysis. This information gives also a hint related to the uncertainty of the derived nocturnal fluxes.

*We appreciate the insights. As indicated in the text, on average across all analyzed FLUXNET sites, latent heat flux is measured in 60 % of all nighttime intervals, whereas gap-filling is required in the remaining 40 %.*

*An alternative to gap-filled fluxes would be to estimate a mean hourly NWL rate from the non-gap-filled observations and obtain total sums by multiplying the mean by the total number of nighttime hours. However, this has its own disadvantages. In any case, results are rather similar with both options.*

Page 4, Line 8: for night-time data?

*We expanded the text. The energy balance correction is applied to both daytime and nighttime data. It uses only half hours with timestamps between 22:00–02:30 and 10:00–14:30. See full details at https://fluxnet.fluxdata.org/data/fluxnet2015-dataset/data-processing/*

Page 4, Line 14-15: move this sentence to the acknowledgements, even though appreciated by myself.

*Ok.*

Page 5, Lines 4ff: this section should be improved by quantitative uncertainty values.

*We expanded the text and modified Fig. 2 to include information about the uncertainty of the NWL estimates from the FLUXNET data.*

Page 6, Line 14: '. . .across sites cannot easily be explained by annual average. . ..'

*We modified the text.*

Page 7, Line 1: can you be sure that EC data are reliable under 'snowy and windy conditions'? EC assumption might not be fulfilled, sonic data are often disturbed under such conditions.

*Our intention here is to point out that conditions at these specific sites are in general snowier and windier than at other sites. We expanded the text to clarify and address this concern.*

Page 10, lines 1ff: for EC estimates no uncertainty is considered. How large are the uncertainties related to the fluxes under consideration?

*We now include uncertainty information within Figure 2 and the corresponding text.*

Page 11, lines 5ff: here it is correctly said that nocturnal measurements can be affected by low turbulence conditions. But nocturnal fluxes are not treated by the energy-balance correction, as also correctly said before. In the discussion part, also the uncertainty of EC data should be discussed.

*We now also refer to the uncertainty of NWL from EC data here.*

Figure 1, caption: should include the site name.

*We added the site name to the caption.*

Figure 2: caption to be extended. What exactly is show? Always consider that reader often concentrate on the figures of a paper only and thus need more information. In addition, in c), the colors of the tiny dots are difficult to distinguish with normal page size. But I also fear, this is not a 'spatial distribution' but rather a 'distribution of sites with . . ..'

*We modified Figure 2 and the caption as well.*

[revised manuscript text omitted]

---

## Author Response (AR2)

**Response to editor**

*Dear authors,*

*Thanks very much for the thorough revision and for using all our comments to further improve the manuscript. Both referees are happy with the revision, but one of them requested some final technical corrections, which I would like you to consider before publication. Could you also consider the following final remarks from my side?*

*1) It is still not made very clear that NWL and sap flow are not necessarily the same thing. If sap flow serves predominantly capacitance re-filling, there can be nocturnal sap flow without any NWL. For example, on P 13 L5-15, NWL and sap flow are used almost interchangeably without clarification of this potential difference.*

We appreciate the comment. We now acknowledge this explicitly in P13 L5–15.

*2) Thank you for the much more detailed description of how to obtain the underlying data, very helpful! I disagree, though, that the scripts used for data selection, analysis and plotting are not relevant. First of all, if someone had the idea to repeat your analysis but change one step or plot a different set of variables, they might be able to save tremendous amounts of time by using your scripts instead of re-coding everything again. Secondly, if someone fails to reproduce your plots using the same data, it would be much easier and faster to find out where things deviate if your scripts were available. I hope you agree that providing your scripts along with the data would tremendously improve the utility and impact of your paper in the future.*

The link to the scripts is now also provided in: "Data and code availability".

**Response to reviewers**

**Reviewer 1:**

*Very well-adjusted manuscript according to the reviewers' suggestions. Thanks for this valuable analysis!*

Thank you.

*Page 2, line 17: the sentence might by reformulated such as: '...latent heat flux and might both occur for example within the same hour, resulting in difficulties to disentangle them ...'*

We reformulated the sentence as suggested.

*Page 3, line 4: '...that nocturnal adjustment of stomatal conductance is an actively controlled process,...' conductance itself is not the process but the state.*

Thanks for the clarification. We reformulated the sentence as follows: "Meanwhile, new evidence suggests that nocturnal stomatal conductance is actively controlled, and that it is not equivalent to minimum conductance (Duursma et al., 2019)".

*Page 3, line 22: '...the stark difference in spatial resolution.' Really 'stark'?*

*We deleted the word "stark".*

**Reviewer 2:**

*The authors appear to have done a good and thorough job responding to comments from the reviewers.*

*Thank you.*

[revised manuscript text omitted]